# Facial and Emotion Recognition Deficits in Myasthenia Gravis

**DOI:** 10.3390/healthcare12161582

**Published:** 2024-08-09

**Authors:** Maddalen García-Sanchoyerto, Monika Salgueiro, Javiera Ortega, Alicia Aurora Rodríguez, Pamela Parada-Fernández, Imanol Amayra

**Affiliations:** 1Neuro-e-Motion Research Team, Department of Psychology, Faculty of Health Sciences, University of Deusto, 48007 Bilbao, Spain; aliciarodriguez.b@deusto.es (A.A.R.); imanol.amayra@deusto.es (I.A.); 2Department of Clinical and Health Psychology and Research Methodology, Faculty of Psychology, University of the Basque Country UPV/EHU, 20018 Donostia, Spain; monika.salgueiro@ehu.eus; 3Centro Investigaciones de Psicología y Psicopedagogía [CIPP], Facultad de Psicología y Psicopedagogía, Pontificia Universidad Católica Argentina, Ciudad Autónoma de Buenos Aires 1107, Argentina; javiera_ortega@uca.edu.ar; 4Consejo Nacional de Investigaciones Científicas y Técnicas (CONICET), Godoy Cruz 2290, Ciudad Autónoma de Buenos Aires 1425, Argentina; 5Child and Adolescent Psychiatric and Psychology Unit, IMQ AMSA, 48010 Bilbao, Spain; pamela.parada@deusto.es; 6Departamento de Psicología, Facultad Ciencias de la Salud, Universidad Europea del Atlántico, 39011 Santander, Spain

**Keywords:** myasthenia gravis, facial emotion recognition, facial recognition, anxious–depressive symptoms

## Abstract

Myasthenia gravis (MG) is a neuromuscular disease of autoimmune etiology and chronic evolution. In addition to the muscle weakness and fatigue that characterize MG, in some studies patients show an inferior performance in cognitive tasks and difficulties in recognizing basic emotions from facial expressions. However, it remains unclear if these difficulties are due to anxious–depressive symptoms that these patients present or related to cognitive abilities, such as facial recognition. This study had a descriptive cross-sectional design with a sample of 92 participants, 52 patients with MG and 40 healthy controls. The data collection protocol included measures to assess recognition of facial expressions (BRFT), facial emotional expression (FEEL), and levels of anxiety and depression (HADS). The MG group had worse performance than the control group in recognizing “fear” (*p* = 0.001; *r* = 0.344), “happiness” (*p* = 0.000; *r* = 0.580), “disgust” (*p* = 0.000; *r* = 0.399), “surprise” (*p* = 0.000; *r* = 0.602), and “anger” (*p* = 0.007; *r* = 0.284). Likewise, the MG group also underperformed in facial recognition (*p* = 0.001; *r* = 0.338). These difficulties were not related to their levels of anxiety and depression. Alterations were observed both in the recognition of facial emotions and in facial recognition, without being mediated by emotional variables. These difficulties can influence the interpersonal interaction of patients with MG.

## 1. Introduction

Myasthenia gravis (MG) is a chronic autoimmune neuromuscular disease. MG affects the production of antibodies that bind to acetylcholine receptors or molecules on the postsynaptic membrane, leading to fatigability and muscle weakness [1,2].

Even though MG is considered a rare disease, it is the most common disease among conditions of the neuromuscular junction. The prevalence of MG ranges from 150 to 200 cases per million inhabitants, but its incidence is increasing in recent years [3,4]. MG can appear at any age and affect both genders equally. However, in women it is usually diagnosed before the age of 40 and in men after the age of 50 [5].

Based on the classification of the Myasthenia Gravis Foundation of America (MGFA) [6], the disease can be divided according to severity and the muscle groups affected. MG can be classified as ocular, axial/spinal, or bulbar. In 50% of cases, patients with MG begin with ocular muscle weakness, manifesting symptoms such as ptosis and double vision or diplopia. However, in most of these cases, the affectation ends up spreading to other muscle groups. In these cases, the most common symptoms are bulbar symptoms (dysarthria and dysphagia), facial weakness, cervical muscle weakness, limb weakness, and dyspnea [1,5,7,8]. These symptoms make it difficult to carry out daily life activities, such as eating or taking care of personal hygiene, and limit other functions, such as driving or performing physical activities. Likewise, some patients exhibit speech impairments and loss of facial expression [7,9], which negatively impact their interpersonal relationships. As a consequence, they present a notable deterioration in their quality of life related to the social domain [10]. This construct is linked to quantitative factors associated with interpersonal and social relationships. In this scenario, patients with MG are forced to give up their social life due to the symptoms they present [8]. Moreover, social functioning is also associated with functional variables, in a qualitative sense, linked to the person’s social resources [11], such as the ability to express and recognize their own emotions and those of others [12]. Facial emotion recognition (FER) plays a main role in recognizing other’s emotions in patients with MG, and impairment of this function has been observed [13].

FER is an ability that allows the interpretation of emotions through facial expression [14,15]. Ekman and Friesen [14] defended the biological origin of six basic emotions (happiness, sadness, fear, surprise, anger, and disgust), which can be recognized universally without the influence of cultural or gender differences. However, the influence of some factors such as age or emotional states on FER can be observed. Regarding age, elderly individuals without illness seem to have more difficulties in identifying emotions expressed through a face compared to young adults [16]. These difficulties are even more pronounced in the recognition of unpleasant emotions such as sadness, anger, or fear [17]. As to emotional states, differences in emotion recognition and reaction time are highlighted in people with higher levels of anxious–depressive symptomatology [15,18]. Similarly, the role of cognitive ability and years of education in FER has been observed, given that FER involves different executive functions, such as attention, perception, memory, and decision-making [19]. Therefore, difficulties in FER may be linked to alterations in these cognitive functions. 

According to the functional cognitive model proposed by Bruce and Young [20], the FER process is divided into three stages: visual processing, structural encoding of facial features, and decoding of facial expression. To that purpose, two brain pathways of FER are described: one for decoding face-specific emotion and the other related to the perception of facial features. The development of FER is linked with efficiency in face encoding [21]. It involves an underlying perceptual mechanism of face recognition, which requires a basic structural representation of faces based on visuospatial scanning [22].

Facial processing can be holistic and/or analytic [23]. Holistic perceptual analysis considers the global features of the face, whereas analytic perceptual analysis considers a set of geometric features of the face [24,25]. It is generally hypothesized that face recognition and FER are composed of the combination of both processing models, but it is emphasized that face identification is a more holistic process than object recognition [26,27]. However, in some pathologies, it has been detailed that faces are processed as a separate set of features, making it hard to correctly interpretate both faces and emotions [28,29]. Among the pathologies that present alteration in FER, psychosis [30], autism spectrum disorders [31,32], and some dementias [33] are considered. Difficulties in identifying emotions, particularly those with negative connotations, are observed in these disorders.

These difficulties have also been described in patients with neuromuscular diseases such as myotonic dystrophies [34], Duchenne and Becker muscular dystrophies [35], and amyotrophic lateral sclerosis [36], in which the motor problems of the disease are associated with delayed emotion recognition [37]. Moreover, being an autoimmune pathology, these difficulties have also been reported in other autoimmune diseases such as multiple sclerosis [38]. In this case, statistically significant deficits were identified for those with negative connotations, specifically, anger, sadness, and fear [38,39]. 

In the case of MG, possible difficulties in recognizing some basic emotions through facial expressions have been described [13]. However, unlike in other pathologies, there are no studies that collected the type of facial recognition processing of patients with MG or the effect of anxious–depressive symptomatology on this ability. Studies in MG are limited to describing a worse cognitive performance in visuoperceptive tests compared to healthy subjects [39]. 

Given that FER deficits negatively impact social and therapeutic interactions, affecting the quality of life of affected individuals [10], the present study aims to extend the scarce literature available in patients with MG. The main objective was to analyze the existence of alterations in facial recognition and in FER in patients with MG, controlling for the effect of anxious–depressive symptomatology. We also aimed to observe to what extent cognitive processes present in facial recognition performance, particularly visuospatial working memory, influence FER in these patients. Finally, we sought to analyze the influence of the clinical variables of MG on facial recognition and facial expression recognition. For that purpose, the different subtypes of MG (ocular, bulbar, or axial) were taken into account with the aim of testing whether there are differences in the skills observed according to the muscle group affected.

## 2. Materials and Methods

### 2.1. Sample

A cross-sectional study with a convenience sample was conducted. The study sample consisted of 92 participants, aged 18 to 75, classified into 2 groups. The clinical group (*n* = 52) consisted of patients from the Neurology Department of the Hospital de Cruces, Bilbao, Spain, with a diagnosis of MG. The healthy subjects were recruited from Spain as part of the control group (*n* = 40) and gender–age matched as a voluntary participation. The sociodemographic and clinical data are shown in Table 1. There are no significant differences between the clinical group and the control group in sex χ^2^(1) = 2.14, *p* = 0.093; age, U = 872.000, *p* = 0.185; or education, U = 806.500, *p* = 0.063. The clinical group was divided into five subgroups based on the classification of the MGFA [3] diagnosed by a neurologist of the Hospital de Cruces: MG Ocular (I) (*n* = 15); MG Generalized Mild (II), extremities and/or axial (*n* = 13) and bulbar (*n* = 7); and MG Generalized Moderate (III), extremities and/or axial (*n* = 2) and bulbar (*n* = 5).

The inclusion criteria for the clinical group were (a) to have a diagnosis of MG established by a neurologist; (b) to be of legal age; (c) to be literate; (d) to have signed the informed consent prior to participation. The inclusion criteria for the control group were (a) to not have a diagnosis of MG; (b) to be of legal age; (c) to be literate; (d) to have signed the informed consent prior to participation.

Exclusion criteria for both groups were (a) presenting a psychiatric or neurological diagnosis other than MG; (b) sensory or motor disorders that may impede psychological testing.

### 2.2. Instruments

After collecting sociodemographic and clinical information through a semi-structured interview, the evaluation protocol used included the following variables: facial recognition, FER, and anxious and depressive symptomatology.

#### 2.2.1. Benton Facial Recognition Test (BFRT)

BFRT [40] is a test designed to measure visual perceptual skills through a face recognition task. In this test, the subject is presented with a series of images of faces under different conditions (different lighting, side view or partial views of the face) and must identify the target face among the different faces shown. The BFRT showed good psychometric properties (α = 0.57; r = 0.60) [41].

#### 2.2.2. Facially Expressed Emotion Labelling Test (FEEL)

FEEL measures a person’s ability to recognize basic emotions through faces [42]. This test presents 42 images of faces of anonymous people, expressing one of the six basic emotions (fear, happiness, surprise, disgust, sadness, and anger), presented through a computer screen for 300 ms. The subject must recognize the expressed emotion. The FEEL has shown good psychometric properties in terms of reliability (α = 0.77) and internal consistency (α = 0.82) in different clinical samples [43,44].

#### 2.2.3. Hospital Anxiety and Depression Scale (HADS)

HADSwas used [45] (Spanish version [46]) in order to measure anxiety and depression. This scale is composed of 14 items divided into two subscales, 7 of which correspond to the anxiety subscale and the other 7 to the depression subscale. The HADS takes into account the cognitive and affective dimensions, leaving aside the somatic aspects. This test showed adequate internal consistency (α = 0.90) [46].

### 2.3. Procedure

Data collection was performed individually at the Cruces Hospital facilities by a psychologist trained in the application of standardized neuropsychological assessment tests. 

It was specified that participation would be voluntary and without financial remuneration. After signing the informed consent, the evaluation began with the mentioned tests and the application of a sociodemographic questionnaire to collect personal and clinical data. The evaluations were each conducted for a duration of one hour per participant and all the assessments were conducted under similar environmental conditions.

Finally, it is stated that this study complied with the criteria of the Psychologist’s Code of Ethics, ensured compliance with the international standards proposed in the Declaration of Helsinki, and was approved by the Research Ethics Committee of the University of Deusto (ETK-54/21-22).

### 2.4. Statistical Analysis

This study used SPSS (Statistical Package for Social Sciences) version 28.0 to perform the analyses. The normal distribution of the sample was tested using the Kolmogorov–Smirnov test. The direct scores were converted into z-scores to carry out the analyses.

In order to analyze the differences between groups with respect to sociodemographic and clinical data, the Chi-square statistic and the Mann–Whitney U test were used for categorical and quantitative variables, respectively. To compare differences in facial recognition and facial expression recognition, the Mann–Whitney U test was used. To calculate the effect size, the r coefficient and Kramer’s V were considered as appropriate. To observe the correlations between the facial recognition test and the facial expressions test, Spearman’s rho coefficient was calculated. The simple linear regression test, with z-scores, was also included to find out to what degree facial recognition can predict the emotional recognition variables.

A multivariate analysis of covariance (MANCOVA) was also carried out, using z-scores, to analyze the influence of anxious–depressive symptomatology, age, and education on the differences found in facial expression recognition and facial recognition. Finally, the Kruskal–Wallis test and the Bonferroni post hoc test were used for intergroup differences. The level of significance was established with a value of *p* < 0.05.

## 3. Results

### 3.1. Differences in Recognition between Clinical and Control Group 

First, differences in facial and emotion recognition between the MG and control groups were explored. The results for each group can be found in detail in Table 2. Statistically significant differences show a low performance in the MG group compared to the control group in facial recognition and in the FER in the emotions of “fear”, “happiness”, “disgust”, “surprise”, and “anger”. No significant differences were observed in the emotion “sadness” and in the total FEEL score. Similarly, no differences were observed in the reaction time between the two groups, except for the emotion “fear”, which the reaction time was longer in the group with MG. 

To determine whether there is a relationship between facial recognition and emotion recognition, Spearman’s correlation was used in the MG group. The BRFT scores showed positive correlation with the FEEL in the emotions of “happiness” (rho = 0.302; *p* = 0.030), “surprise” (rho = 0.299; *p* = 0.031), and “anger” (rho = 0.301; *p* = 0.030). A simple linear regression model with input method was calculated to find out to what extent the degree of facial recognition, analyzed through the BRFT test, influences FER. The results show significant regressions for the emotions “happiness” (F = 4.24, R2 = 0.078, β = 0.28, *p* = 0.045), “surprise” (F = 4.93, R2 = 0.09, β = 0.30, *p* = 0.031), and “anger” (F = 9.10, R2 = 0.11, β = 0.33, *p* = 0.019).

### 3.2. Impact of Anxiety and Depression in Recognition

On the other hand, according to the HADS test, significantly higher levels of anxious and depressive symptomatology in the different subscales were observed in the MG group compared to clinical group.

Given the differences found in facial recognition and FER, a MANCOVA analysis was carried out to control for the influence of anxious–depressive symptomatology. Age and education have been included as covariates. Table 3 shows the results comparing both groups, MG and control. Results showed that differences were maintained between the groups in facial recognition and emotion recognition, except for the emotion anger (F = 3.45, *p* = 0.067). The effects of corticosteroids on these abilities were tested by means of the Mann–Whitney U test, which did not give any significant results.

### 3.3. Differences between MG Subgroups

To determine the differences in facial recognition and facial expression recognition in the different MG subgroups (ocular, axial, and bulbar), the Kruskal–Wallis H test and Bonferroni post hoc test were performed. The results were statistically significant in facial recognition (H = 9.726, *p* = 0.008) between the ocular (M = 44.60) and bulbar (M = 48.45) subtypes (*p* = 0.011), with participants with bulbar symptomatology performing better.

In regard to FER, differences were significant in the recognition of the “fear” emotion (H = 7.566, *p* = 0.023) between the ocular (M = 2.20) and bulbar (M = 4.05) subtypes (*p* = 0.022); in the “happiness” emotion (H = 10.751, *p* = 0.005) between the ocular (M = 2.60) and bulbar (M = 4.95) (*p* = 0.006) subtypes; in the “surprise” emotion (H = 6.693, *p* = 0.035) between the ocular and bulbar subtypes (M = 4.07 and M = 5.23, respectively; *p* = 0.001); and in the “disgust” emotion (H = 13.379, *p* = 0.001) between the axial and bulbar subtypes (M = 4.07 and M = 4.53, respectively; *p* = 0.047). Thus, the bulbar subgroup obtained a superior performance in the recognition of these emotions.

Moreover, concerning reaction time (RT), significant differences were observed between the ocular and axial subtypes in the emotion “happiness” (H = 7.767, *p* = 0.021), “surprise” (H = 7.601, *p* = 0.022), and “sadness” (H = 8.289, *p* = 0.016). In the Kruskal–Wallis Bonferroni post hoc test, statistically significant differences were found in the RT of “happiness” (*p* = 0.016) between the ocular (M = 2010.91) and axial subtypes (M = 3009.97); the RT of the emotion “surprise” (*p* = 0.024) between the ocular (M = 2752.85) and axial (M = 3250.69) subtypes; and the RT of the emotion “sadness” (*p* = 0.013) between the ocular (M = 3250.32) and axial (M = 4281.73) subtypes. Overall, higher RT was obtained in patients with the axial subtype.

## 4. Discussion

The purpose of the present study was to examine the extent to which facial decoding can explain the difficulties that patients with MG may present in FER once the effect of variables such as age, level of education, or anxious–depressive symptomatology in patients was controlled.

In this study, the MG group showed a lower performance compared with the control group in the recognition of the “fear”, “surprise”, “disgust”, “happiness”, and “anger” emotions. Therefore, MG participants demonstrate poorer performance, consistent with previous research indicating difficulties in recognizing emotions via facial expressions in patients with MG [13], which reported differences between the two groups in the “fear” and “surprise” emotions.

In seeking to learn more about variables that may influence FER, the multivariate analysis of covariance showed that by controlling for the effects of anxious–depressive symptomatology, age, and education, statistically significant differences between the clinical group and the control group were maintained in all emotions except for “anger”. This effect is also described in other studies with the general population in which participants with higher levels of anxiety showed more accurate recognition of faces expressing anger compared to participants with low anxiety [47]. Similarly, previous studies highlight the influence of age on the recognition of negative emotions [17]. Despite this, in this study age, education, and affective variables do not seem to be related to the difficulties encountered in FER in patients with MG.

Following on from the objectives, given the assumption that correct facial discrimination is required to adequately process facial emotions [21], facial recognition skills were analyzed in MG patients. The results revealed significant differences in facial recognition between the clinical group and the control group, with participants in the MG group showing inferior performance. The correlation analyses showed a positive correlation between facial recognition and FER in the emotions of “happiness”, “surprise”, and “anger”. However, through the simple linear regression test, it was shown that, in this sample, facial recognition only explains between 7% and 10% of the scores in the mentioned emotions. 

Given that the relationship between FER and visuospatial face recognition was modest, it seems necessary to analyze the intervention of other variables. Previous studies have described memory impairments, spatial and non-spatial learning, and alterations in visuospatial tasks in patients with MG [48]. In other autoimmune diseases such as multiple sclerosis, the relationship between cognitive function decline and poorer performance in facial expression recognition remains independent of psychological or disease-related factors [49,50,51].

For this reason, the described difficulties in FER and face recognition could be linked to previously described cognitive impairments in patients with MG, more explicitly to difficulties related to face rotation processes related to visuospatial working memory. Patients who presented greater difficulty in the BFRT test in relating the image of the same person in a frontal perspective and in an angular or profile perspective showed more errors in the recognition of some facial emotions in frontal perspective in the FEEL. This error may have to do with the skills required in combining analytical and holistic processing. It is likely that there is an impairment in the integration of the analytical information of the most characteristic features of the face with the holistic view as an integrated whole [23,24,25,26]. On the other hand, these visuospatial deficits do not only occur in face recognition but also in other contexts, such as in spatial orientation in general [48].

The loss of gray matter in MG patients in the fusiform gyrus and cingulate gyrus [48], areas involved in FER and face recognition, could also explain the impairments in these skills. However, we do not rule out the possibility of impairments in other brain regions [52] that could explain difficulties in FER, for example, the amygdala, insula, or globus pallidus, as has been observed in multiple sclerosis and other neurologically affected diseases [53,54]. 

Another theory that may account for difficulties in FER is the “embodied recognition” conception. This raises a fundamental requirement for individuals to engage in sensorimotor activities within their own bodies to achieve proficient processing and identification of emotions [55]. This theoretical framework draws on the concept that experiential engagement with one’s body movements and sensory experiences is integral to the nuanced understanding of emotional expressions [34].

Previous investigations have already established a compelling association between emotion recognition capabilities and motor impairments in neurodegenerative disorders like Huntington’s disease and Parkinson’s disease [56,57]. These findings highlighted the intricate interplay between motor function and the intricate neural networks responsible for decoding facial expressions. This association is also described in neuromuscular diseases such as myotonic dystrophy [34] and Duchenne and Becker muscular dystrophy [35]. 

In the context of recent research, a study specifically conducted with patients diagnosed with MG revealed a discernible reduction in the expression of emotions such as anger, fear, and happiness among these patients [58]. Therefore, patients with MG may present alterations in emotional recognition due to the interruption in the integration of bodily and sensory experiences in the emotional recognition process. However, it is important to consider the hypothesis that suggests the involvement of the condition in the neural pathways responsible for proper facial recognition. Furthermore, the presence of characteristic symptoms of myasthenia gravis, such as diplopia and ptosis, could have had a significant influence on the development and test results. These clinical manifestations, affecting visual function and ocular motor skills [1,6], could have hindered the accurate perception and interpretation of facial expressions during the evaluation of emotional recognition. This observation may support the rationale behind interpreting the data wherein diminished performance is observed within the group with ocular MG. It is possible that some patients with bulbar myasthenia gravis may experience less significant ocular involvement, which could influence their ability to recognize visual facial expressions more effectively compared to those with ocular myasthenia gravis. In the bulbar MG group, although the disease may be more severe overall, its impact may be more related to muscle weakness in areas such as chewing, swallowing, and speech, rather than directly affecting visual perception and facial expression recognition. In addition, although no influence of corticosteroid use has been observed, the dose and duration used in each case and other treatments received have not been monitored. Therefore, it is crucial to consider both neuromuscular deficits and specific symptoms of myasthenia gravis when interpreting the results of studies on emotional recognition in this population. 

Despite the relevance of the evidence and conclusions drawn, this study had some limitations. First, the sample size used is limited due to the complexity involved in recruiting participants in studies with rare diseases, such as MG. A second limitation was that the number of participants in the control group was lower because we wanted to match age and sex with the MG control group, and this was not possible for all participants. In addition, it would be useful to control clinical variables (antibodies or/and daily symptomatology of each patient) in order to know the differences between the different subtypes of myasthenia gravis more concretely. Moreover, although in this study corticosteroids had no effect on cognitive performance, it would be advisable to monitor the dosage and durability of corticosteroid use, as well as the effects of other drugs. The use of the FEEL may be another limitation, because the analysis focuses on identifying static facial expressions rather than dynamic stimuli. In addition, the scarcity of existing articles related to emotion recognition in patients with MG has limited the conclusions of this study. Finally, as mentioned throughout the study, cognitive functions are involved in cognitive recognition, so it would be interesting to take them into account in future studies.

Therefore, it is recommended that future research include other cognitive aspects related to emotions which have not been collected so far in patients with MG, such as social cognition. Likewise, further research is suggested along the lines of the current findings, with the detailed differences studied taking into account the different types of MG. The results of this study imply difficulties that these patients may have in their interpersonal relationships [8] and clinicians are encouraged to take facial recognition into account in their clinical assessments and treatment planning, as called for in previous research that considers social cognition [59], in order to provide comprehensive person-centered care.

## 5. Conclusions

In conclusion, this study has identified possible significant alterations in emotion and facial recognition among patients with MG, which may impact their interpersonal interactions and overall quality of life. These findings emphasize the need for broader consideration of MG’s effects beyond its typical symptoms. Addressing these challenges comprehensively can enhance support strategies and interventions to improve social integration and well-being in these people. It is therefore important for social services and researchers to take these findings into account when designing interventions and for future research in this area.

## Figures and Tables

**Table 1 healthcare-12-01582-t001:** Sociodemographic and clinical characteristics of the total sample.

Variable		Clinical Group (*n* = 52) *M (SD)*/*n* (%)	Control Group (*n* = 40) *M (SD)*/*n* (%)	*p*-Value
Gender	Women	35 (67.3%)	20 (50%)	0.093
	Men	17 (32.7%)	20 (50%)	
Age (years)		53.90 (13.10)	50.78 (10.83)	0.185
Education (years)		13.62 (4.00)	12.08 (2.97)	0.063
Years with the disease		10.40 (11.54)	-	
Corticoids	Yes	32 (61.5%)	-	
	No	20 (38.5%)	-	
MGFA Type	MG Ocular (I)	15 (28.8%)	-	
	MG Generalized Mild (II)			
	Extremities and/or axial	13 (25.0%)	-	
	b. Bulbar	17 (32.7%)	-	
	MG Generalized Moderate (III)			
	Extremities and/or axial	2 (3.8%)	-	
	b. Bulbar	5 (9.6%)	-	

Note. *n* = number of participants; *M* = mean; *SD* = standard deviation.

**Table 2 healthcare-12-01582-t002:** Differences in facial recognition, facial expression recognition, and anxious–depressive symptomatology between the clinical group and the control group.

Variable	Clinical Group (*n* = 52) Mdn (Range)	Control Group (*n* = 40) Mdn (Range)	*U*	*Z*	*p*	*r*
FEEL						
FEEL total	54.50 (58)	60.50 (56)	1147.500	0.847	0.397	0.089
Fear	2.00 (6)	5.00 (7)	1451.000	3.282	0.001 *	0.344
Happiness	2.00 (5)	7.00 (2)	1681.000	5.513	0.000 **	0.580
Anger	6.00 (7)	7.00 (5)	1358.000	2.712	0.007 *	0.284
Surprise	3.00 (5)	7.00 (3)	1735.000	5.750	0.000 **	0.602
Disgust	5.00 (7)	6.50 (7)	1509.500	3.809	0.000 **	0.399
Sadness	5.00 (7)	6.00 (6)	1215.500	1.439	0.150	0.151
RT Fear	2812.50 (5179.14)	2594.00 (2507.00)	782.500	−2.028	0.043 *	0.213
RT Happiness	1902.43 (10,236.57)	1868.71 (3589.43)	1021.500	−0.146	0.884	0.015
RT Anger	2571.57 (7571.57)	2172.86 (4278.71)	814.500	−1.776	0.076	0.186
RT Surprise	2377.64 (5174.14)	2231.86 (2413.71)	925.500	−0.902	0.367	0.094
RT Disgust	3193.86 (7247.57)	2743.36 (4071.57)	820.500	−1.729	0.084	0.181
RT Sadness	2875.22 (7542.57)	2334.50 (3363.00)	855.500	−1.753	0.146	0.184
BFRT	47.00 (17)	49.50 (14)	1447.500	3.222	0.001 *	.338
HADS						
Total	12.00 (27)	6.00 (27)	645.000	−3.113	0.002 *	0.327
Depression	5.00 (17)	1.00 (11)	602.500	−3.477	0.001 *	0.364
Anxiety	6.50 (16)	4.50 (17)	740.500	−2.366	0.018 *	0.248

Note: *n* = number of participants; Mdn = median; *U* = Mann–Whitney U test; *Z* = z-scores; * *p* ≤ 0.05; ** *p* ≤ 0.001; *r* = r coefficient (effect size).

**Table 3 healthcare-12-01582-t003:** MANCOVA facial recognition and facial expression recognition controlling for the effect of anxious–depressive symptomatology, age, and education.

Variable	Clinical Group (*n* = 52) *M* (*SD*)	Control Group (*n* = 40) *M* (*SD*)	*F*	*p*	*η* ^2^ _p_
FEEL					
Fear	−0.28 (0.96)	0.42 (0.92)	7.485	0.008 *	0.080
Happiness	−0.54 (1.01)	0.72 (0.25)	52.021	<0.001 **	0.377
Anger	0.02 (0.89)	0.32 (0.81)	1.910	0.171	0.022
Surprise	−0.52 (0.95)	0.74 (0.36)	60.808	<0.001 **	0.414
Disgust	−0.21(0.75)	0.41 (1.02)	8.191	0.005 *	0.087
TR Fear	−0.21 (0.59)	−0.46 (0.38)	4.738	0.032 *	0.052
BFRT	−0.20 (1.02)	0.48 (0.76)	10.287	0.002 *	0.107

Note: *n* = number of participants; *M* = mean; *SD* = standard deviation; *F* = MANCOVA test; * *p* ≤ 0.05; ** *p* ≤ 0.001; *η*^2^_p_ = partial eta squared (effect size).

## Data Availability

The datasets presented in this article are not readily available because they belong to the University of Deusto. Requests to access the datasets should be directed to the corresponding author (Maddalen García-Sanchoyerto).

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
