# Peer review of "Facial and Emotion Recognition Deficits in Myasthenia Gravis"

_healthcare, 2024, doi:10.3390/healthcare12161582_

Round 1
Reviewer 1 Report
Comments and Suggestions for Authors
Thank you for inviting me to review this manuscript. Authors present an original papersl studying the variations in emotion and facial expressions in oatients having myasthenia gravis. In my opinion, authors well prepared the manuscript, and it offers an addition to tge literature. I have some comments that can help improve the manuscript:
1- The abstract lacks numbers. It is better to add some p values.
2- It is better to add p value in the table comparing demographics of patients and controls. This is necessary to see if there is a confounding bias. For example, gender may affect emotions and facial expressions, so if there is a significant difference in gender between cases and controls, this may affect the credibility of the results.
3- How are controls picked? This can benefit from a better explanation.
4- Authors mentioned that having a small sample is a limitation, and explained the cause for that. However, I do not see a cause for a small sample of controls. Why are controls less than cases? This needs explanation.
5- The conclusion can be more elaborated. I suggest adding some future perspecrives.
I beleive this manuscript could be interesting for readers, and offers an addition to the literature.
Good luck
Author Response
First of all, I would like to thank reviewer for her/his helpful suggestions. I am sure that these changes will contribute to the improvement of the paper. The suggested revisions are described below mentioning point-by-point what changes have been made. After each suggestion, the answer is described written in bold.
Thank you for inviting me to review this manuscript. Authors present an original papersl studying the variations in emotion and facial expressions in oatients having myasthenia gravis. In my opinion, authors well prepared the manuscript, and it offers an addition to tge literature. I have some comments that can help improve the manuscript:
1- The abstract lacks numbers. It is better to add some p values.
The p-values have been added to the abstract.
2- It is better to add p value in the table comparing demographics of patients and controls. This is necessary to see if there is a confounding bias. For example, gender may affect emotions and facial expressions, so if there is a significant difference in gender between cases and controls, this may affect the credibility of the results.
The p-values have been included in the table 1 to make them clearer.
3- How are controls picked? This can benefit from a better explanation.
To make the sample recruitment clearer, the following sentence has been added in the sample section (line 121-122): The healthy subjects were recruited from Spain as part of the control group (n = 40), by gender-age matched as a voluntary participation.
4- Authors mentioned that having a small sample is a limitation, and explained the cause for that. However, I do not see a cause for a small sample of controls. Why are controls less than cases? This needs explanation.
In lines 349-351 the following paragraph has been added to explain why there were fewer participants in the control group: “A second limitation was that the number of participants in the control group was lower because we wanted to match age and sex with the MG control group, and this was not possible for all participants.”
5- The conclusion can be more elaborated. I suggest adding some future perspecrives.
A paragraph has been added to the conclusions, line 378-380. “It is therefore important for social services and researchers to take these findings into account when designing interventions and for future research in this area.”
In addition, following the recommendations of other reviewers who have also requested to add implications of the study, has been added a paragraph to the discussion section that determines the extent of this study (line 367-371). “The results of this study imply difficulties that these patients may have in their inter-personal relationships [8] and clinicians are encouraged to take facial recognition into account in their clinical assessments and treatment planning, as called for in previous research that consider social cognition [59], in order to provide comprehensive person-centred care.”
Reviewer 2 Report
Comments and Suggestions for Authors
It could be beneficial to expand the discussion to more explicitly connect how the findings might impact clinical practices or patient care. Discuss the potential for using facial affect recognition as a diagnostic tool or a method to evaluate disease severity or treatment efficacy.
The limitation part could include discussing potential biases, the generalizability of the findings to other populations, or limitations inherent in the technology used for facial recognition.
Author Response
First of all, I would like to thank reviewer for her/his helpful suggestions. I am sure that these changes will contribute to the improvement of the paper. The suggested revisions are described below mentioning point-by-point what changes have been made. After each suggestion, the answer is described written in bold.
It could be beneficial to expand the discussion to more explicitly connect how the findings might impact clinical practices or patient care. Discuss the potential for using facial affect recognition as a diagnostic tool or a method to evaluate disease severity or treatment efficacy.
We agree with this suggestion so a paragraph has been added to the discussion (line 367-371): “The results of this study imply difficulties that these patients may have in their inter-personal relationships [8] and clinicians are encouraged to take facial recognition into account in their clinical assessments and treatment planning, as called for in previous research that consider social cognition [59], in order to provide comprehensive person-centred care.
The limitation part could include discussing potential biases, the generalizability of the findings to other populations, or limitations inherent in the technology used for facial recognition
We agree with the need to explain limitations that may exist at the methodological or technological level. In order to address this appreciation a paragraph has been added to the limitations in line (358- 360): “The use of FEEL test may be another limitation, because the analysis focuses on identifying static facial expressions rather than dynamic stimuli.”
Reviewer 3 Report
Comments and Suggestions for Authors
This article investigated facial recognition and expressions according to the phenotype of MG and the muscle group affected. It's a very interesting work and a novelty in the field of neuromuscular diseases.
The manuscript is well-written and fluid. The methodology is well explained and the results well presented.
As minor comments :
- I would like to ask authors if the cognitive functions were assessed in the included patients and if findings were adjusted according to this parameter.
- Authors might add some practical implications related to these findings.
Author Response
First of all, I would like to thank reviewer for her/his helpful suggestions. I am sure that these changes will contribute to the improvement of the paper. The suggested revisions are described below mentioning point-by-point what changes have been made. After each suggestion, the answer is described written in bold.
This article investigated facial recognition and expressions according to the phenotype of MG and the muscle group affected. It's a very interesting work and a novelty in the field of neuromuscular diseases.
The manuscript is well-written and fluid. The methodology is well explained and the results well presented.
As minor comments :
- I would like to ask authors if the cognitive functions were assessed in the included patients and if findings were adjusted according to this parameter.
In this case, the cognitive functions of the participants were not assessed, due to the difficulty of carrying out this type of study with people with fatigue who require a brief assessment. It has been included in limitations line 362-363: “Finally, as mentioned throughout the study, cognitive functions are involved in cognitive recognition, so it would be interesting to take them into account in future studies.”
- Authors might add some practical implications related to these findings.
We agree with this suggestion, which is shared by several reviewers, so it has been added a paragraph to the discussion (line 367-371): “The results of this study imply difficulties that these patients may have in their interpersonal relationships [8] and clinicians are encouraged to take facial recognition into account in their clinical assessments and treatment planning, as called for in previous research that consider social cognition [59], in order to provide comprehensive person-centred care.”